# Computer Vision and Machine Learning-Based Gait Pattern Recognition for Flat Fall Prediction

**DOI:** 10.3390/s22207960

**Published:** 2022-10-19

**Authors:** Biao Chen, Chaoyang Chen, Jie Hu, Zain Sayeed, Jin Qi, Hussein F. Darwiche, Bryan E. Little, Shenna Lou, Muhammad Darwish, Christopher Foote, Carlos Palacio-Lascano

**Affiliations:** 1State Key Laboratory of Mechanical System and Vibration, Shanghai Jiao Tong University, Shanghai 200240, China; 2Orthopaedic Surgery and Sports Medicine, Detroit Medical Center, Detroit, MI 48201, USA; 3South Texas Health System—McAllen Department of Trauma, McAllen, TX 78503, USA

**Keywords:** machine learning, gait, pattern recognition, convolutional neural network, support vector machine, long short-time memory, k nearest neighbor, fall recognition

## Abstract

Background: Gait recognition has been applied in the prediction of the probability of elderly flat ground fall, functional evaluation during rehabilitation, and the training of patients with lower extremity motor dysfunction. Gait distinguishing between seemingly similar kinematic patterns associated with different pathological entities is a challenge for the clinician. How to realize automatic identification and judgment of abnormal gait is a significant challenge in clinical practice. The long-term goal of our study is to develop a gait recognition computer vision system using artificial intelligence (AI) and machine learning (ML) computing. This study aims to find an optimal ML algorithm using computer vision techniques and measure variables from lower limbs to classify gait patterns in healthy people. The purpose of this study is to determine the feasibility of computer vision and machine learning (ML) computing in discriminating different gait patterns associated with flat-ground falls. Methods: We used the Kinect^®^ Motion system to capture the spatiotemporal gait data from seven healthy subjects in three walking trials, including normal gait, pelvic-obliquity-gait, and knee-hyperextension-gait walking. Four different classification methods including convolutional neural network (CNN), support vector machine (SVM), K-nearest neighbors (KNN), and long short-term memory (LSTM) neural networks were used to automatically classify three gait patterns. Overall, 750 sets of data were collected, and the dataset was divided into 80% for algorithm training and 20% for evaluation. Results: The SVM and KNN had a higher accuracy than CNN and LSTM. The SVM (94.9 ± 3.36%) had the highest accuracy in the classification of gait patterns, followed by KNN (94.0 ± 4.22%). The accuracy of CNN was 87.6 ± 7.50% and that of LSTM 83.6 ± 5.35%. Conclusions: This study revealed that the proposed AI machine learning (ML) techniques can be used to design gait biometric systems and machine vision for gait pattern recognition. Potentially, this method can be used to remotely evaluate elderly patients and help clinicians make decisions regarding disposition, follow-up, and treatment.

## 1. Introduction

As the population ages, the emergence of diseases such as lower extremity diseases or motor nerve dysfunction have increased leading to an increased rate of elderly adults falling on flat ground. Approximately 32% of community-dwelling elderly adults over 75 years of age will fall at least once during a one-year interval and 24% of these individuals will sustain serious injuries [1,2]. In the United Kingdom (UK), the medical costs related to falls are substantial; fall-related injuries in adults greater than 60 years have been reported to cost more than £981 million pounds per year [3]. Total healthcare spending for elderly falls ranged from $48 million in Alaska to $4.4 billion in California. Medicare spending attributable to older adult falls ranged from $22 million in Alaska to $3 billion in Florida, as reported in 2014. The lifetime medical costs of fall-related injuries ranged from $68 million in Vermont to $2.8 billion in Florida [3]. As such, falling has become a costly problem in the growing elderly population [1,2,4].

For that reason, the detection and recognition of fall risk has been growing due to the implementation of safety measures [5] in high-risk work environments, hospitals, and nursing homes [6].

A person’s pattern of walking can be understood by gait analysis. Gait and balance functions decline through the course of disorders including stroke, dementia, Parkinson’s disease, arthritis, and others [7,8,9]. Gait can serve as a marker of changes in physical status and fall risk [10]. The gait of the human body refers to the behavioral characteristics of the lower limbs of the human body in the process of upright walking. A normal human gait cycle usually needs to meet the characteristics of natural, coordination of the legs, labor saving, and periodicity. Abnormal gait can develop before the human body falls. Numerous possibilities may cause an abnormal gait. In the field of medical rehabilitation, identification and evaluation of abnormal gait patterns significantly guide lower limb training regimens and flat ground falls prevention strategies.

By monitoring the gait patterns of elderly patients, proper preventive measures can be recommended to reduce the risk of flat ground falls. Human vision may not accurately recognize or quantify the changes in the gait pattern. Therefore, automatic gait recognition using computer vision has become a hot research topic in the biomechanics and healthcare literature in recent years [10,11,12].

Computer-vision technology is used to acquire gait kinematics information, including angles, velocity, and acceleration of the joints based on Kinect skeletal tracking sequences [12,13]. The gait analysis involves many interdependent measures that can be difficult to interpret due to a vast amount of data and their inter-relations, and a significant amount of labor is required in off-line data analysis [14]. This has led to the development of machine learning (ML) for data analysis, and ML has been used for gait analysis [13,14]. Dolatabadi et al. [13] used two ML approaches, an instance-based discriminative classifier (k-nearest neighbor) and a dynamical generative classifier (Gaussian process latent variable model) to distinguish between healthy and pathological gaits with an accuracy of up to 96% when walking at a fast pace. Ortells et al. [15] developed vision-based gait-impairment analysis to distinguish a normal gait from seven impaired gaits for aided diagnosis. Zakaria et al. [16] classified Autism Spectrum Disorder (ASD) children’s gait from normal gait by a depth camera and found the accuracy of the support vector machine (SVM) classifier was 98.67% and the Naive Bayes classifier had an accuracy of 99.66%. Chen et al. [17] and Xu et al. [18] classified Parkinsonian gait classification using monocular video imaging techniques and kernel-based principal component analysis (PCA) [17,18]. These studies demonstrate the advancement of sensor technology and its capacity to collect kinematic and electrophysiological information during walking, which has greatly promoted the development of automated gait recognition technology. However, there remains a lack of real-time computer vision monitoring systems for geriatric gait monitoring for fall-risk warning or home-based remote monitoring. Computer vision using ML computing techniques for flat falls has not been fully understood nor investigated.

ML techniques have been studied for gaiting pattern recognition among different disorders. ML can be used for real-time signal processing and instant output command signals [19,20,21]. The pathway includes collecting data for ML model training and then using it for real-time signal monitoring and the process for commanding signal output for control. Our long-term goal is to develop a real-time ML-based computer vision system for geriatric gait analysis that is capable of predicting flat ground fall risk. We plan to collect gait data among elderly patients with a flat ground fall history to build ML models. This study aimed to design a cost-effective gait-recognition system for the elderly population using different ML algorithms and identify the optimal ML method for building a computer camera monitoring system.

This paper proposed the design of a human gait recognition system using the Azure Kinect camera (Microsoft Inc., Redmond, Seattle, WA, USA). Compared with traditional motion capture equipment, the Kinect camera has the characteristics of easy integration, no need for large-scale data acquisition equipment, and a user-friendly experience. We hypothesized that different gait patterns can be classified and recognized using computer vision and ML techniques. Using the Kinect skeleton tracking technology, the spatial information of the key points of the human skeleton can be accurately obtained, processed through the algorithm, and converted into the joint angle during walking. Unlike other studies that only collect ankle joint information [12], footstep information [18], trunk tilt angle [22], or data from a public database [23], we collected all joint kinetic data available for ML processing. Moreover, previous studies detected the whole fall event [12,18,22,23,24], while our study focused on detecting abnormal gaits (pelvic obliquity gait and knee hyperextension gait) aiming at the prevention of falls. Four automatic classification algorithms were used to classify human gait patterns, including Convolutional Neural Network (CNN), Support Vector Machine (SVM), Long Short-Time Memory (LSTM), and K-Nearest Neighbors (KNN) [24,25,26,27]. The purpose was to determine the optimal classifier for flat fall gait recognition.

## 2. Materials and Methods

### 2.1. Participants and Experimental Devices

Seven healthy subjects (aged 23 to 29 years), including 3 males and 4 females, were recruited in this study. None of the subjects had any history of neurological or musculoskeletal disorders. This study was approved by the Institutional Review Board (IRB) of Shanghai Jiao Tong University (I2022210I). All procedures were performed according to the Declaration of Helsinki. Each subject signed an informed consent form before the start of the experiment.

The equipment used in this study included an Azure Kinect DK integrated camera (Microsoft Inc., Redmond, Seattle, WA, USA), an h/p/cosmos treadmill, and a laptop. The Azure Kinect DK integrated camera was used to photograph and record the key point information of the subject’s angle during the experiment. The camera was placed on the lateral side of a participant aiming at the central axis of the frame. The whole body of a participant was captured for recording. Participants were video recorded during the gait task. These tasks allowed us to visually detect features of 3 gait patterns including normal gait (NG), pelvic obliquity (PO) gait (pelvic-hiking gait) [28,29], and knee hyperextension (HK) gait (trunk forward tilt gaits). The h/p/cosmos treadmill was used to control the experimenter’s gait walking speed, and the laptop was used to run software programs and ML algorithms. The software used for walking motion information acquisition included Visual Studio 2019 (Microsoft Inc., Redmond, Seattle, WA, USA), OpenCV (Version 3.4.10, OpenCV Team, Palo Alto, CA, USA), and MATLAB (version 2018, MathWorks, Natick, MA, USA). The ML algorithms were implemented using the PyCharm (Python version 3.7, Boston, MA 02210, USA) and PyTorch (version 1.11, San Francisco, CA, USA) software.

### 2.2. Experimental Procedures

Before the start of the experiment, the participant was given instruction to perform 3 patterns of walking gait and allowed to practice on the treadmill for 3 min. Before the formal experiment, each subject conducted a testing experiment, including practicing 3 designated gaits (Figure 1) to test equipment connectivity and system setup. During the experiment, the subjects were asked to look forward as much as possible and maintain a steady gait on the treadmill.

In this experiment, the subjects walked on the h/p/cosmos treadmill with the 3 different gait patterns shown in Figure 1. The 3 types of gaits are normal gait (NG) during normal walking, abnormal gait patterns including pelvic obliquity (PO) gait, and knee hyperextension (KH) gait patterns of the right lower limb. The abnormal gaits were generated for the right lower limb only. Pelvic obliquity gait refers to first the torso being lifted to the left when walking, then lifting the pelvis and stepping out with the right leg. During the knee hyperextension gait, the trunk was tilted forward slightly while the right knee was lifted to step out during walking. These gait patterns occur at the critical moment of a fall and therefore can be used for fall detection [30].

During the experiment, the speed of the treadmill was set to run at a constant speed of 0.1 m/s. A subject walked on the treadmill at the same speed for 1 min. For each subject, each gait pattern was recorded 5 times to obtain 5 sets of valid experimental datasets; a total of 15 sets of valid experimental datasets were obtained for offline data analysis.

### 2.3. Kinect Skeleton Tracking Technique

Microsoft Azure Kinect SDK (software development kit) camera and a human skeleton recognition and tracking software toolkit (Azure Kinect Body Tracking SDK) were used in this study. The subject’s motion was tracked in the camera’s depth image field of view with 32 joint point angle readings recorded and saved in the computer while tracking the target. The spatial coordinates and skeleton joint nodes of the upper limb and lower limb, spine, shoulder, and hip are shown in Figure 2. In this study, the joint points of the lower limbs were extracted for gait pattern recognition. According to the joint points of the lower limbs, the angle of joint flexion and extension was calculated and then used as the featured value of gait recognition.

The skeleton nodes used in the calculation of the lower limb joint angle included right clavicle point A (Clavicle_Right), left clavicle point B (Clavicle_Left), pelvis point C (Pelvis), right hip joint point D (Hip_Right), left hip joint point E (Hip_Left), right knee point F (Knee_Right), left knee point G (Knee_Left), right ankle point H (Ankle_Right), left ankle point I (Ankle_Left) (Figure 2).

The joint angles used in this experiment include the hip joint flexion and extension angle, hip joint abduction and adduction angle, and knee joint angle. Their definitions are defined as follows:

Hip joint flexion-extension angle: the angle between the thigh vector line (the line connecting D–F points) and trunk line (the line connecting A–D points) on the plane on which these two lines intersect for the right hip joint, and the angle between the E–G line and B–E line for the left hip joint.

Hip joint abduction-adduction angle: the angle between the thigh vector line (the line connecting D–F points) and the pelvic line (the line connecting C–D points) on the plane on which these two lines intersect for the right hip joint, and the angle between the E–G line and C–E line for the left hip joint.

Knee joint angle: the angle between the thigh vector line (the line connecting D–F points) and the leg vector line (the line connecting F–H points) on the plane on which these two lines intersect for the right knee joint, and the angle between the E–G line and G–I line for the left knee joint.

Taking the calculation of the right knee joint as an example, when the coordinates of joint point D were set as (*x*_1_, *y*_1_, *z*_1_), joint point F as (*x*_2_, *y*_2_, *z*_2_), and joint point H as (*x*_3_, *y*_3_, *z*_3_); then the right thigh vector $\vec{DF}$ was expressed as (*X*_1_, *Y*_1_, *Z*_1_), and the right leg vector $\vec{FH}$ as (*X*_2_, *Y*_2_, *Z*_2_), in which *X*_1_
*= x*_1_
*− x*_2_, *Y*_1_
*= y*_1_ − *y*_2_, *Z*_1_
*= z*_1_
*− z*_2_, *X*_2_
*= x*_2_
*− x*_3_, *Y*_2_
*= y*_2_
*− y*_3_, *Z*_2_
*= z*_2_
*− z*_3_. Thus, the right knee angle $\alpha \;\left(\vec{DF},\vec{FH}\right)$ was calculated using the following equation:(1)$$\alpha \left(\vec{DF},\vec{FH}\right)=\cos^{-1}\left(\frac{X_{1}*X_{2}+Y_{1}*Y_{2}+Z_{1}*Z_{2}}{\sqrt{X_{1}^{2}+Y_{1}^{2}+Z_{1}^{2}}*\sqrt{X_{2}^{2}+Y_{2}^{2}+Z_{2}^{2}}}\right)$$

The definition and calculation of six joint angles used in this gait pattern recognition study are shown in Table 1.

The joint angle calculation algorithms were implemented using Visual Studio software (Version 2019, Microsoft Inc., Redmond, Seattle, WA, USA) and C++ language. The frame rate (FPS) of the Kinect camera was set up as 5 fps, in which 5 images per second of the point of the skeleton node positions were captured and processed. The featured vector *F_θ_* was calculated using six lower limb joint angles of each image of the skeleton node frame. The featured *F_θ_* is expressed as the following equation:(2)$$F_{\theta }=\left\{\theta_{1},\theta_{2},\theta_{3},\theta_{4},\theta_{5},\theta_{6}\right\}$$

Participant walking speed was set at 0.1 m/s by following the treadmill moving speed. Because a participant was able to complete at least one gait cycle of the walk within 3 s, a time window of 3 s was used to segment the recorded data into datasets. A total of 2100 sets ([60/3] × 5 × 3 × 7)((60 s/3 s of processing time window) × 5 times of walking on the treadmill per gait pattern × 3 gait patterns × 7 participants) of valid datasets were collected. Each dataset contained 90 (6 × 5 × 3) (6 joint angle readings × 5 frames per second (FPS) × 3 s of the time window for data processing) feature values and their corresponding gait labels of normal, pelvic obliquity gait, and knee hyperextension gait. The data segmentation was performed using MATLAB software.

### 2.4. Machine Learning-Based Classifiers

In this study, four classifiers were used to classify gait data, including the convolutional neural network (CNN), long short-term memory (LSTM) neural network, support vector machine (SVM), and K-nearest neighbors (KNN) classifier. CNN and LSTM algorithms were implemented using PyCharm and PyTorch. SVM and KNN algorithms were implemented using MATLAB and Toolbox.

#### 2.4.1. Convolutional Neural Network (CNN) Classifier

The convolutional neural network with two 2-dimensional convolutional layers was used in this experiment. The CNN is a type of deep neural network classifier that has been used in image classification and recognition [31,32]. It can recognize features directly from the data instead of manually extracting features. Within the CNN structure in this study, the data was put through several layers with different tasks [33,34]. The input was imported into the convolution layer where a spatial filter was applied to the inputs in the form of a window of weights. This spatial filter moved vertically and horizontally throughout the inputs to generate the output of the convolution layer, which was rectified by a rectified linear unit (ReLU) and then exported into the pooling layer. The results of the pooling layer were put through a fully connected layer yielding the classification [35,36,37,38,39,40]. This approach has been used in medical applications where datasets are of a small size [40], demonstrating success in medical image recognition [41]. During the training phase, an epoch was defined as a full training cycle using the whole training set, and a mini-batch size was a subset of the training set used to update the weights and calculate the gradient descent loss function.

The structure of the proposed CNN is shown in Figure 3. The input of the CNN was the gait angles derived from Kinect; two 2D-convolutional layers were used in the model, and the output of the model was the classification result of the gait pattern. In the first convolutional layer, the number of filters was 16, kernel size 5, padding 2, and stride 1. For the max pooling in the first layer, the kernel size was 2, padding 0, and stride 2. In the second layer, the number of filters was 32, kernel size 5, padding 2, and stride 1. For the max-pooling in the second layer, the kernel size was 2, padding 0, and stride 2.

The loss function of the model was the cross-entropy loss function, which can be calculated by the following formula:(3)$$loss=\left(x,class\right)=-\log\left(\frac{\exp\left(x\left[class\right]\right)}{\sum_{j}\exp\left(x\left[j\right]\right)}\right)\;=-x\left[class\right]+\log\left(\sum_{j}\exp\left(x\left[j\right]\right)\right)$$
where *x* is the input, class is the index of the target, and *j* is the number of classes.

The optimizer of the model was Adam. The training was completed after 500 epochs with a mini-batch size of 16 images when the learning rate was fixed and set to 0.001.

#### 2.4.2. Support Vector Machine (SVM) Classifier

In this study, the support vector machine (SVM) model was applied as one of the classifiers and a Bayesian optimizer was used to optimize the model. SVM [42] uses different and high dimensional features of the datasets to assign a label to a data point based on their location in the data space. SVM as a supervised learning algorithm has many applications [18], such as face detection [43] and bioinformatics [44]. It is an optimal hyperplane in an N-dimensional space as a decision boundary to separate datasets into two or more classes. [45]. In this study, we used SVM and the Bayesian optimizer to identify an optimal classifier for gait pattern recognition.

#### 2.4.3. K-Nearest Neighbors (KNN) Classifier

In this paper, we used an automatically optimized KNN model; a Bayesian optimizer was selected as an optimizer and the distance measurement method was implemented using the Euclidean metric method.

KNN (K-nearest neighbors) is another supervised classification algorithm that has no explicit training phase [46]. The principle of KNN predicts the category of a new value according to the category of the K points closest to it. When using KNN, many parameters need to be optimized, including the number of adjacent points, distance metrics, and distance weights. KNN classification is based on a distance metric between a data point to be classified and already known data points from a training set [47,48]. KNN assigns that label to a data point which is shared by the majority of the K nearest surrounding data points of the training set. The setting of the K value affects the accuracy of the model, so the K value in the proposed method varies in the optimization process, and the K value varies between 1–150.

#### 2.4.4. Long Short-Term Memory (LSTM) Neural Network

LSTM is a special recurrent neural network (RNN) structure that can memorize long- and short-term information. Recurrent neural networks (RNNs) are a class of neural networks that are naturally suited to processing time-series data and other sequential data. The traditional RNN network usually produces long-term dependencies after multiple calculations of network nodes, whereas the LSTM network avoids this phenomenon through the design of the structure [49,50].

Because gait actions are often closely related to the moments before and after a certain moment, we used a bidirectional LSTM (Bi-LSTM) network for classification. The architecture of the proposed model was shown in Figure 4. The model was composed of two Bi-LSTM layers and a fully connected (FC) layer. The bi-LSTM layers could process the input information both in forward and backward directions. The output of the current moment is not only affected by the previous state but may also be affected by the future state.

The loss function of the model was a cross-entropy loss function, which can be calculated by Formula (3). The optimizer of the model was Adam. The training was completed after 500 epochs with a mini-batch size of 20 images when the learning rate was fixed and set to 0.0005.

### 2.5. Statistical Analysis

One-way ANOVA with PostHoc LSD was used to determine the statistical difference of the average range of motion (ROM) of a joint between different gait patterns, as well as the average accuracy of gait pattern recognition between ML algorithms. A Pearson’s chi-square or Bonferroni correction test was performed to determine the difference in the accuracy of each gait pattern recognition among the four individual machine learning algorithms for different gait patterns. SPSS software (Version 28, IBM, Armonk, NY, USA) was used for statistical analysis. A *p*-value smaller than 0.05 was considered to be statistically significant.

## 3. Results

### 3.1. Characteristics of Lower Limb Joint Angles in Three Gait Patterns

A Microsoft Azure Kinect SDK camera and the Human Skeleton Recognition and Tracking software toolkits (Azure Kinect Body Tracking SDK) captured the joint angle during the subject’s walking on a treadmill. Figure 3 shows the joint angles of a subject’s right hip and knee within 3 to 9 s among different gait pattern groups.

The average hip joint flexion-extension angle was 32.6 ± 3.8 degrees in the normal gait group, 14.1 ± 3.0 degrees in the PO group, and 16.0 ± 3.5 degrees in the KH group (Figure 5a). The ROM of the right hip joint of the normal gait (NG) was larger than the ROM of the PO and KH gait patterns (ANOVA, PostHoc LSD, *p* < 0.001). The ROM of the right hip joint of the KH gait was larger than that of the PO gait (ANOVA, PostHoc LSD, *p* < 0.001).

The average ROM of hip joint abduction was 18.9 ± 4.9 degrees in the normal gait group, 17.1 ± 4.6 degrees in the PO group, and 9.5 ± 2.6 degrees in the KH group (Figure 5b). The abduction ROM of the right hip joint of normal gait (NG) was larger than that of the PO and KH gait patterns (ANOVA, PostHoc LSD, *p* < 0.001). The ROM of the right hip joint of the PO gait was larger than that of the KH gait (ANOVA, PostHoc LSD, *p* < 0.001).

The average ROM of knee joint flexion-extension was 48.7 ± 5.4 degrees in the normal gait group, 18.8 ± 5.5 degrees in the PO group, and 27.6 ± 3.8 degrees in the KH group (Figure 5c). The ROM of the right knee joint of normal gait (NG) was larger than that of the PO and KH gait patterns (ANOVA, PostHoc LSD, *p* < 0.001). The ROM of the right knee joint of the KH gait was larger than that of the PO gait (ANOVA, PostHoc LSD, *p* < 0.001).

### 3.2. Comparison of the Model Training Process

Eighty percent of the data was used for training, and 20 percent of the data was used for validation testing to verify the accuracy of the trained model. Figure 6, Figure 7, Figure 8 and Figure 9 are the plots showing the loss value over the training iterations of 4 machine learning algorithms. CNN model training demonstrated a typical curve characteristic of gradual convergence with a minimum loss after 100 training iterations with loss values of 0.032 and 0.006 at the 200th iteration (Figure 6). SVM model training demonstrated a rapid convergence after 200 training iterations with a minimum loss value of 0.0125 (Figure 7). KNN model training appeared to have a convergence after 200 training iterations with a minimum classification loss value close to 0.06 (Figure 8). LSTM model training had a convergence at the 200th training iteration with a loss value of 0.25, but the loss value increased with the increase of iterations, demonstrating overfitting [51] (Figure 9). SVM had the shortest training iteration to archive the minimum error, followed by CNN, KNN, and LSTM. CNN had the smallest minimum error.

To compare the reasoning speed of the model, we also tested the runtime of the four proposed models, and the result is shown in Table 2. The training epochs for CNN and LSTM were 500, and for SVM and KNN the training epochs were 200. All four proposed methods were running on the same computer. The average runtime of CNN was 20.17 ± 0.99 s, while the average runtime of LSTM was 32.77 ± 1.32 s. The training epochs for SVM and KNN were smaller than for CNN and LSTM, but the runtime of SVM and KNN was longer than that of CNN and LSTM. The average runtime for SVM was 714.71 ± 58.14 s, and the average runtime for KNN was 316.57 ± 11.41 s. Table 2 shows the best K value for the KNN model.

### 3.3. Comparison of Classification Accuracy of the Four ML Algorithms

In this experiment, we used a total of four different classification models to recognize gait movements. The recognition accuracy of the four classification algorithms is shown in Figure 10. Overall, the algorithm with the highest recognition accuracy was the SVM algorithm, with an average recognition accuracy of 94.9 ± 3.36%. The accuracy of the KNN algorithm was 94.0 ± 4.22%, CNN 87.6 ± 7.50%, and LSTM 83.6 ± 5.35% among participants. LSTM had a lower accuracy (Pearson’s chi-square test, *p* = 0.031) (Table 3).

The classification accuracy of the two traditional machine learning algorithms was significantly higher than that of the algorithm using the deep learning architecture, which has a stronger relationship with our data processing this time. This time, we identified the key points of the skeleton in the human gait image and calculated the joint angles of the lower limbs, which greatly reduced the feature dimension of the input data, so the network architectures such as CNN and LSTM lost their advantages. Machine learning algorithms such as SVM and LSTM have relatively simple structures and obvious advantages in dealing with such low-dimensional features. For the time window, 3 s is used to separate the data, which ensures that each time window covers the complete gait action, thus achieving a high recognition accuracy.

Figure 11 shows the confusion matrix results of four classification methods in gait recognition, and Table 4, Table 5, Table 6 and Table 7 provide detailed information on the confusion matrix and corresponding precision and recall of four machine learning algorithms.

### 3.4. Outcomes of the Chi-Square Test

Statistical analysis demonstrated that there was not a statistical difference between the 4 ML methods in predicting normal gait patterns (chi-square, Bonferroni correction test, *p* > 0.7). CNN had higher accuracy in predicting the PH gait pattern (*p* = 0.007) while KNN has a lower accuracy (*p* = 0.001). CNN had higher accuracy in predicting the BK gait (*p* = 0.04) while LSTM had a lower accuracy (*p* = 0.000) (Table 8).

## 4. Discussion

### 4.1. Application of Computer Vision Camera in Gait Analysis

Azure Kinect is a cutting-edge spatial computing developer kit with sophisticated computer vision, advanced AI sensors, and a range of powerful SDKs that can pair real-time depth sensor data with AI-driven insights. Current development techniques target prevention and mitigation of potential patient accidents and injuries in healthcare environments with predictive alerts. The Azure Kinect camera also consists of an RGB camera and an IR camera. The color camera can offer a resolution of 3840 × 2160 px at 30 Hz. The IR camera can have a resolution of 1 MP 1024 × 1024 pixels. It has been successfully used and validated in human gait analysis [52,53,54]. The RGB-D sensor system can estimate features of gait patterns in pathological individuals and differences between disorders. The advances and popularity of low cost RGB-D sensors have enabled us to acquire depth information of objects [55,56,57]. The Kinect sensor also utilizes the time-of-flight (ToF) principle. Time-of-flight is a method for measuring the distance between a sensor and an object based on the time difference between the emission of a signal and its return to the sensor, after being reflected by the object [58]. Both the RGB and IR cameras support different fields of view. The Azure Kinect also has an IMU sensor, consisting of a triaxial accelerometer and gyroscope, which can estimate its position in space. Microsoft offers a Body Tracking SDK for human skeletal tracking using C and C++ programming languages. This SDK tracks a total of 32 main joints of a participant to provide the joint orientations in quaternions representing local coordinate systems and 3D position data [52]. Our study demonstrated that the Microsoft Azure Kinect SDK camera and Human Skeleton Recognition and Tracking software toolkits (Azure Kinect Body Tracking SDK) captured the joint angles as SDK defines them during subject gait analysis on a treadmill. The ML algorithms recognized the three different gait patterns described in this study.

In this study, the hip angles were measured by a computer camera and calculated using computer-vision-marked joint lines (clavicle-point to hip-point to knee-point) defaulted by SDK software based on its 3D coordinate axis. This joint angle measurement is different from the conventional calculation methods with different joint markers. Most of the traditional methods for hip joint angle measurement are based on the angle between the thigh and body trunk. Thus, the gait curves of our study are different from the typical gait curves [59,60] measured by traditional methods [61]. The knee joint calculation method is similar to traditional measuring methods (hip-point to knee-point to ankle-point). Although there is a disparity between joint measurement measures, our results demonstrated that computer vision and ML computing can classify normal gait from PO and KH gaits.

Most of the previous studies demonstrated that Azure and the Body Tracking SDK can recognize the gait patterns and differences in individuals with disorders [52,54,55,59]. Azure Kinect tracking accuracy of the foot marker trajectories is significantly higher than the previous model Kinect v2 overall treadmill velocities. Azure Kinect can efficiently express human poses and gestures [52,62]; however, the newly added joints (clavicles, spine chest) of the Azure Kinect camera achieved a reasonable tracking error [52]. There is an occlusion problem causing the quality of invisible joints to be randomly degraded [62]. It has been suggested that tracking quality can be enhanced by improving hardware and deep-learning (DL)-based skeleton tracking algorithms. Microsoft has started to develop a new body-tracking SDK for the Azure Kinect using DL and convolutional neural networks (CNN) machine learning (ML) algorithms [52]. However, there is a paucity of research reports on gait analysis using Azure Kinect or other computer-vision-based ML and DL techniques [63]. Our study suggested that the Azure camera and Human Skeleton Recognition and Tracking software toolkits can effectively discriminant the gait patterns using machine learning algorithms.

### 4.2. Machine Learning for Gait Analysis

Compared with computer vision-based, most ML-based gait analysis adopt IMU-based sensor systems [63,64,65,66,67]. The ML techniques used in IMU-based gait analysis include decision tree (DT) [68], linear discriminant analysis (LDA) [69], k-nearest neighbors (k-NN) [70], support vector machine (SVM) [67], CNN [71,72], random forest (RF) [61,66,73], and LSTM [49]. The efficiency of ML algorithms on Azure Kinect computer vision-based gait analysis is still not clear. The LSTM or CNN model has a generalization ability across different scenes, while the regularization and generalization techniques of CNN [51] and LSTM [74] have been used in image processing [33,34,48,56]. In this study, four different classification methods including convolutional neural network (CNN), support vector machine (SVM), K-nearest neighbors (KNN), and long short-term memory (LSTM) neural network were used to automatically classify three gait patterns. Their efficiency at gait pattern classification was investigated and compared. The SVM (94.9 ± 3.36%) had the highest accuracy in the classification of gait patterns, followed by KNN (94.0 ± 4.22%). The accuracy of CNN was 87.6 ± 7.50% and LSTM 83.6 ± 5.35%. The accuracy of SVM and KNN was higher than that of CNN and LSTM. During model training, CNN had the smallest training loss value followed by SVM (0.0125), KNN (0.03), and LSTM (0.25, then increased) with overfitting. SVM had the shortest training iteration to archive the minimum error, followed by CNN, KNN, and LSTM. CNN had the smallest minimum error. These outcomes demonstrated that LSTM was less efficient in gait pattern classification in this study. The reason for lower LSTM accuracy in gait pattern classification could be that LSTM algorithms are generally designed for larger amounts of data with complex structures. The estimation errors increased over the iterations upon using LSTM in this study, which could be due to overfitting [51]. Overfitting has been reported in the literature, and the estimation errors of the proposed LSTM methods gradually became high over the steps when it is used for constant acceleration motion tracking [74]. SVM and KNN are designed for small-size dataset applications. CNN fits better for medium-sized datasets. The dataset we collected in this study was smaller; thus, SVM and KNN revealed better performance.

Fricke et al. [75] compared three machine learning algorithms (KNN, SVM, and CNN) for the automatic classification of electromyography (EMG) patterns in gait disorders and found that CNN had the highest accuracy (91.9%), while SVM (accuracy 67.6%) and KNN (accuracy 48.7%) had much lower accuracy. Our study demonstrated that the ML approach of our gait pattern recognition using computer vision was similar to Fricke’s EMG-based machine learning process method. However, our results demonstrated that computer vision yields relatively higher accuracy than EMG-based ML for gait pattern recognition.

### 4.3. Epidemiology of Flat Fall

Falling is a common problem in the growing elderly population [4]. Approximately 32% of community-dwelling elderly adults aged ≥75 years fall at least once during a one-year interval and 24% of these individuals sustain serious injuries [1,2]. The medical costs related to falls are substantial [3]. In 2012, there were 24,190 fatal and 3.2 million medically treated non-fatal fall-related injuries. Direct medical costs totaled $616.5 million for fatal and $30.3 billion for non-fatal injuries in 2012 and rose to $637.5 million and $31.3 billion, respectively, in 2015 [76].

The risk factors associated with elderly falls include age-related gait pattern changes, and deficits in the musculoskeletal system, proprioception, and vestibular system [4,77]. Predicting falls remains an elusive task for healthcare providers. For this reason, many researchers and clinicians have used screening tools based on known risk factors in an attempt to identify future “fallers” [78]. Although a history of falls remains the most accurate predictor of future falls [79], it is not useful for the early detection and prevention of first-time falls [80].

Elderly persons presenting with mild cognitive impairments (dementia) or using a walker/crane demonstrate a higher fall rate [81]. Similarly, comparing individuals with gait deficiencies to those with a history of falls suggests that executive function, gait control, and fall risk may be linked [81]. Because most falls occur during locomotion, research has focused on identifying age-related gait pattern changes [82].

### 4.4. The Rationale of Our Gait Pattern Selection for the Study

Hip abduction can compensate for a reduced swing-phase knee flexion angle in those after a stroke. Pelvic obliquity (hip hiking) also facilitates foot clearance with greater energy efficiency [28,29]. This gait is most likely a reflection of an altered motor template occurring after a stroke, which contributes to a decline in gait ability [83,84]. Home-based stroke hemiplegia patients tend to fall easily due to poor toe clearance, which is reported to be one of the causes of falling, although there are many other related factors [85]. The hyperextension gait reflexes a hyperextension moment of the knee joint during walking, which is commonly seen among post-stroke patients with a stiff knee. The stiff-knee gait is a common abnormal gait pattern in patients after stroke characterized by insufficient knee flexion during swing [86,87]. Elder adults can develop abnormal gaits that are similar to pathologic gaits such as frontal gait, hemiparetic gait, Trendelenburg gait, and knee hyperextension gait [88]. Abnormal gaits are usually caused by musculoskeletal and/or neurologic abnormalities, which can be identified in the clinical setting. Prompt diagnosis and appropriate treatment may save an elderly patient from immobility, fall-related injury, loneliness, and depression [89]. Early detection of abnormal gait with related interventions will be critical in general fall prevention. Because both PO and KH gaits are common abnormal gaits among elder adults and patients with neurological disorders, they were selected for this study to develop a computer vision and machine learning-based sensor system aiming at elder adult abnormal gait detection and fall prevention.

### 4.5. Limitations of This Study

The participants of this study were healthy volunteers, who mimicked the pathologic gaits by walking on a treadmill. The efficiency of this computer vision and machine learning-based sensor system in the classification of patients’ gaits in a clinical setting is unclear. Future study should include clinical studies using this sensor system to recognize the differences between flat-faller and non-flat-faller’s gait conditions for gait pattern recognition.

In summary, this paper proposed a gait recognition system using a computer vision Kinect skeleton tracking technology based on machine learning algorithms. Compared with the traditional gait recognition technology, this method has the characteristics of easy construction, rapid calculation, and good user experience. Further study may demonstrate the utility of computer vision and ML processes to detect aberrant gait patterns in clinical settings. Additionally, future studies should focus on the utilization of the Azure–Kinect in assessing pathological gait patterns of postoperative patients undergoing rehabilitation.

## 5. Conclusions

This paper proposes a gait recognition system using computer vision and machine learning-based human motion tracking technology. The recognition accuracy of the system can reach 94.5%, which meets the needs of users; providing support for the use of the proposed AI machine learning techniques to design gait biometric systems and machine vision applications for gait pattern recognition. Potentially, this method can be used to evaluate geriatric gait patterns, predict flat falls, and aid clinicians in decision-making regarding both disposition and rehabilitation regimens.

## Figures and Tables

**Figure 1 sensors-22-07960-f001:**
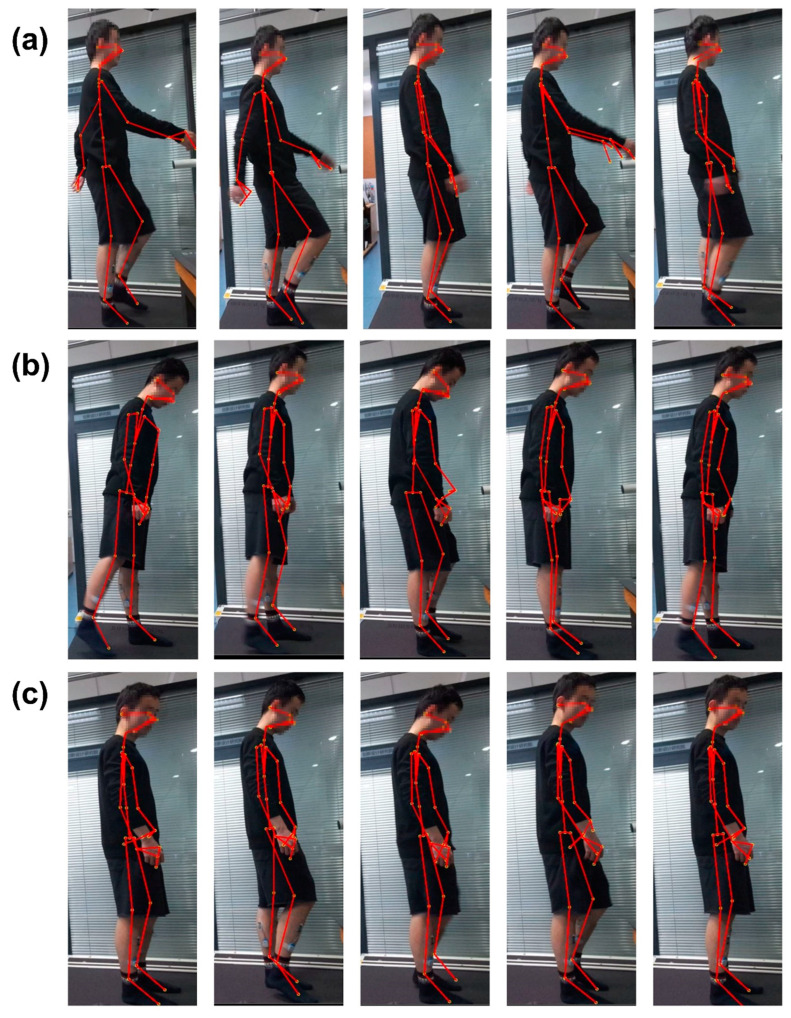
Illustration of gait kinematics information acquisition, including angles and velocity of the joints using Kinect skeletal tracking SDK, Open CV, and Visual Studio software. Kinematics information of 3 gait patterns was obtained for offline data analysis. (**a**) The tracking sequence of normal gait (NG); (**b**) The tracking sequence of pelvic obliquity (PO) gait; (**c**) The tracking sequence of knee hyperextension (KH) gait.

**Figure 2 sensors-22-07960-f002:**
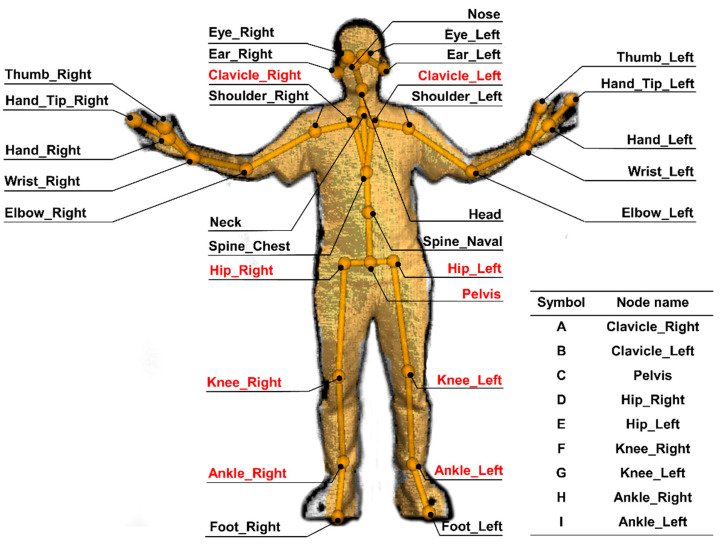
The human joints that can be obtained by the Kinect SDK camera and the Human Skeleton Recognition and Tracking software toolkits. These 32 joint points are automatically generated by Kinect SDK software based on machine-detected motions. Points A-I represent the joints used in this study for joint angle calculations.

**Figure 3 sensors-22-07960-f003:**
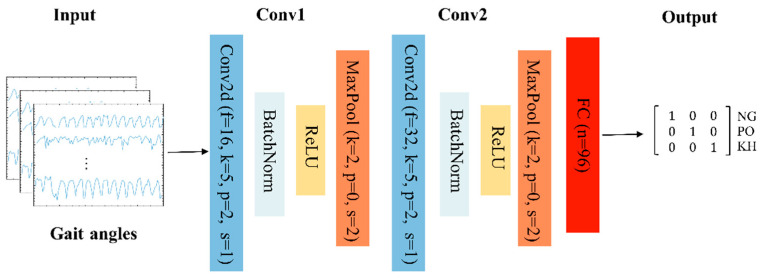
Proposed CNN structure for gait recognition. There are two, 2D-convolutional layers with the following parameters: f = number of filters, k = kernel size, s = stride, p = padding, and n = number of nodes.

**Figure 4 sensors-22-07960-f004:**
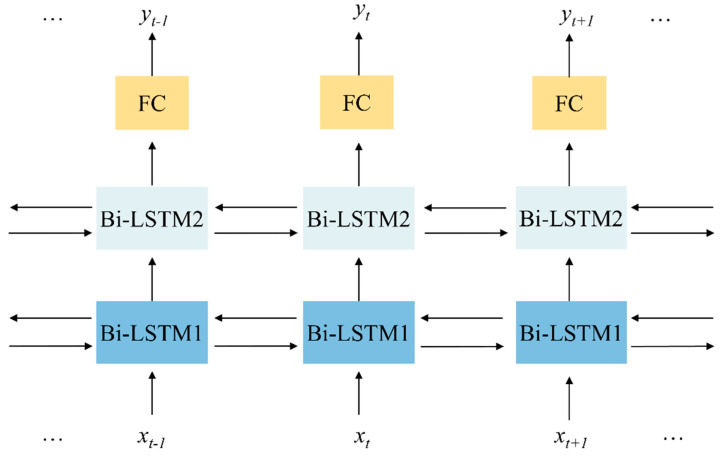
The architecture of the proposed LSTM model.

**Figure 5 sensors-22-07960-f005:**
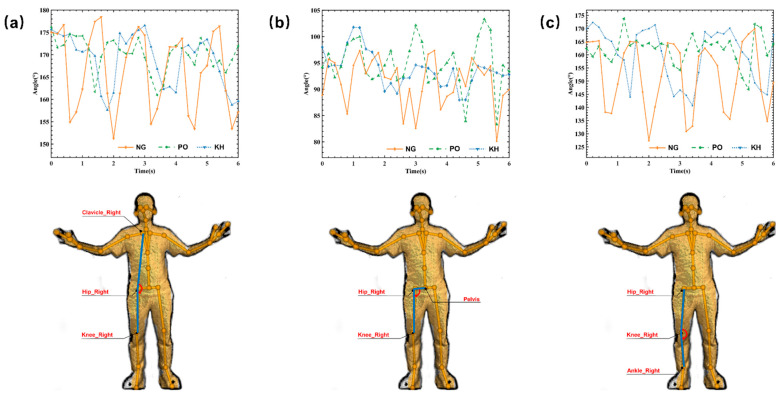
Graph of right lower limb joint angles over time during walking measured by the Human Skeleton Recognition and Tracking software. (**a**) Hip joint flexion-extension angle; (**b**) Hip joint abduction-adduction angle; (**c**) Knee joint angle. NG (normal gait) represents the angles in normal gait, PO (pelvic obliquity) represents the angles during pelvic obliquity gait, and KH (knee hyperextension) represents the angles during knee hyperextension gait. The upper row shows joint angles over time. The lower row demonstrates the measured joint angles (blue lines) in this study.

**Figure 6 sensors-22-07960-f006:**
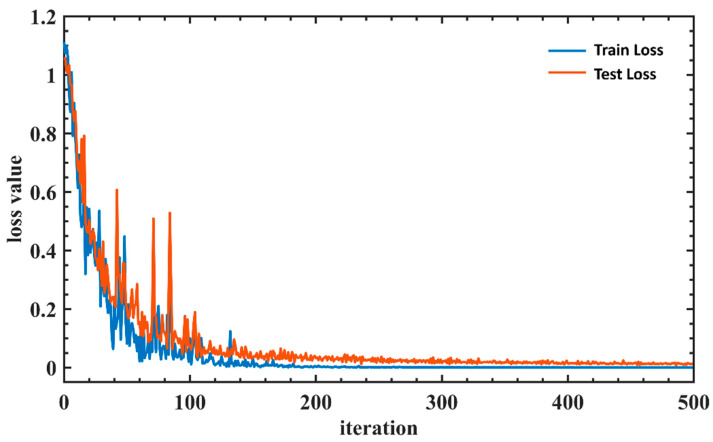
CNN training loss value over iterations. The classic loss curves were found in the CNN model training and model validation. The loss value of the CNN model converges rapidly in the early stage of training. The loss curve plateaus after about 200 iterations, indicating that the model has converged after 200 iterations.

**Figure 7 sensors-22-07960-f007:**
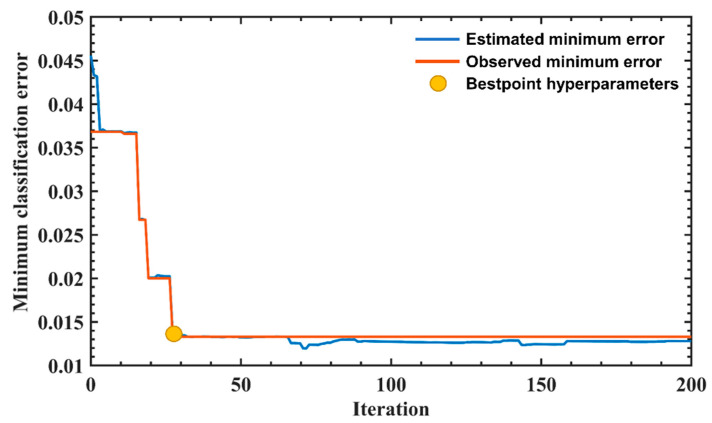
SVM training loss value over iterations. The process of automatic optimization of the SVM model shows the observation error decreases with the increase in iterations. At the 30th iteration, the SVM model has the smallest loss value, indicating that the SVM model achieves the most optimal parameters.

**Figure 8 sensors-22-07960-f008:**
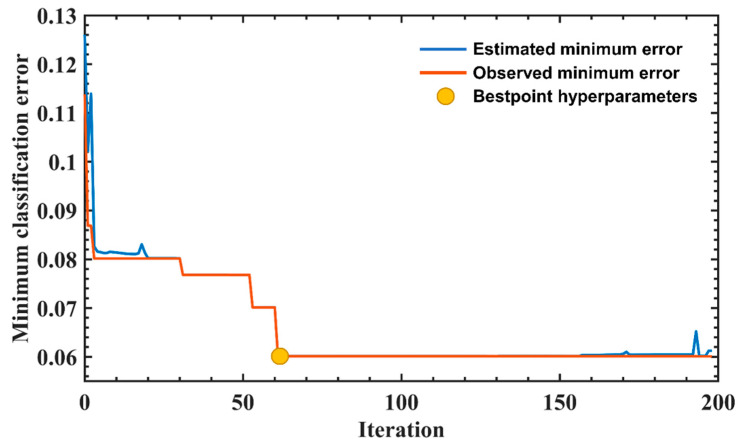
KNN training loss value over iterations. The KNN model yields the smallest classification error around the 60th iteration, indicating that the KNN model has achieved the optimal parameters.

**Figure 9 sensors-22-07960-f009:**
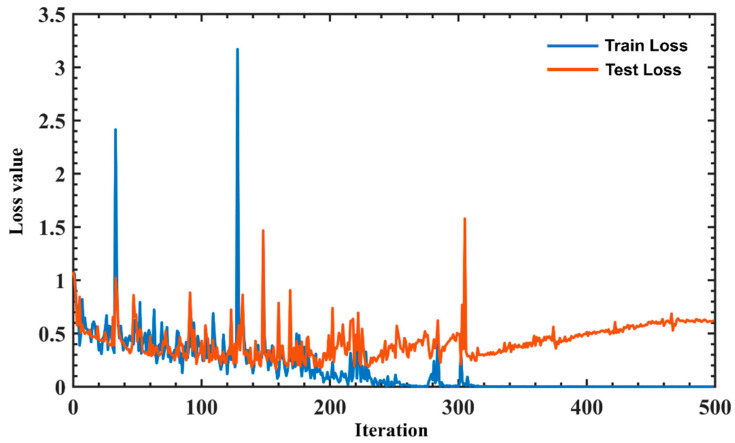
LSTM training loss value over iterations. There is an overfitting event that occurred over the training iterations. There is no convergence over the training iterations; however, the loss value drops over the iterations.

**Figure 10 sensors-22-07960-f010:**
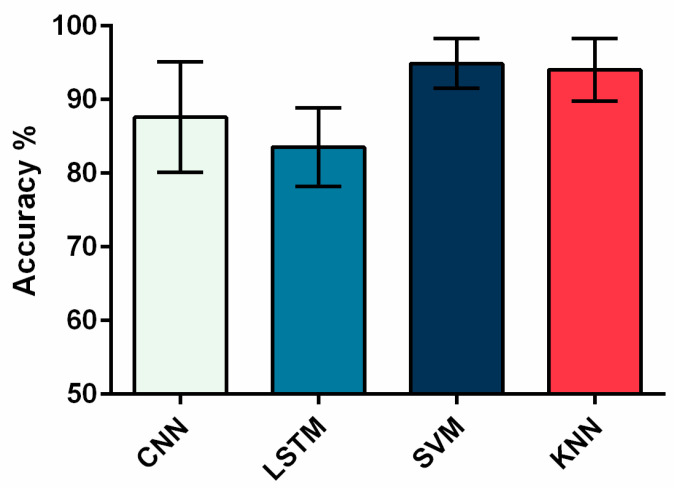
Comparison of the average classification accuracy of different models.

**Figure 11 sensors-22-07960-f011:**
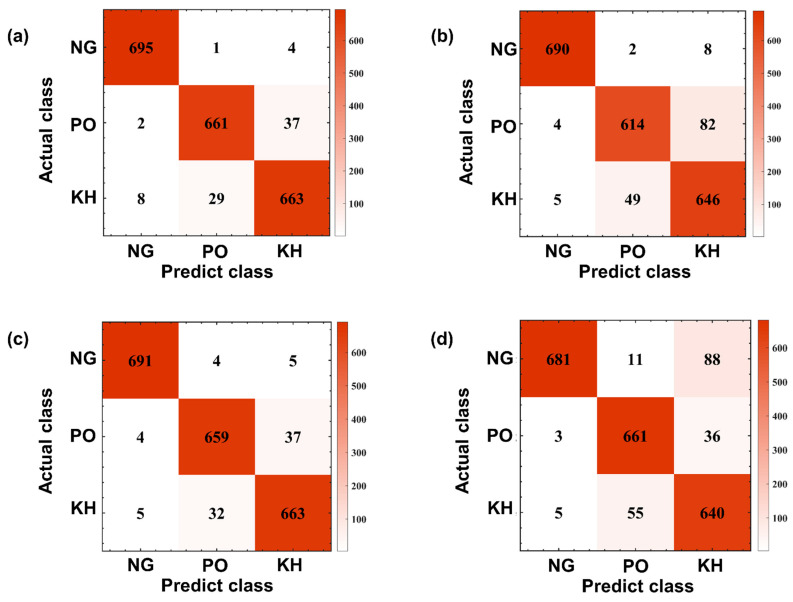
Confusion matrix result of four classification methods: (**a**) CNN; (**b**) LSTM; (**c**) SVM; (**d**) KNN.

**Table 1 sensors-22-07960-t001:** The definition of joint angles used in this study.

No	Symbol	Description
1	$\theta_{1}=\alpha \left(\vec{AD},\vec{DF}\right)$	Left thigh flexion angle
2	$\theta_{2}=\alpha \left(\vec{BE},\vec{EG}\right)$	Right thigh flexion angle
3	$\theta_{3}=\alpha \left(\vec{DF},\vec{CD}\right)$	Left hip joint angle
4	$\theta_{4}=\alpha \left(\vec{EG},\vec{CE}\right)$	Right hip joint angle
5	$\theta_{5}=\alpha \left(\vec{EG},\vec{GI}\right)$	Left knee angle
6	$\theta_{6}=\alpha \left(\vec{DF},\vec{FH}\right)$	Right knee angle

**Table 2 sensors-22-07960-t002:** The runtime of four proposed models.

	CNN	LSTM	SVM	KNN
Subject 1	20.9 s	32.8 s	658 s	300 s (k = 1)
Subject 2	21.5 s	32.7 s	757 s	317 s (k = 10)
Subject 3	19.2 s	33.5 s	663 s	315 s (k = 6)
Subject 4	19.7 s	33.4 s	646 s	334 s (k = 6)
Subject 5	19.4 s	33.5 s	763 s	316 s (k = 2)
Subject 6	19.3 s	29.9 s	728 s	327 s (k = 2)
Subject 7	21.2 s	33.6 s	788 s	307 s (k = 20)
Average	20.17 ± 0.99 s	32.77 ± 1.32 s	714.71 ± 58.14 s	316.57 ± 11.41 s

**Table 3 sensors-22-07960-t003:** Accuracy of seven subjects in the gait recognition experiment.

	CNN	LSTM	SVM	KNN
Subject 1	85.0%	86.6%	92.3%	89.3%
Subject 2	75.0%	71.7%	89.7%	91.3%
Subject 3	98.3%	88.3%	99.3%	99.7%
Subject 4	90.0%	85.0%	95.3%	93.3%
Subject 5	83.3%	81.7%	92.0%	88.7%
Subject 6	85.0%	83.3%	98.0%	97.0%
Subject 7	96.7%	88.3%	97.7%	99.0%
Average	87.6 ± 7.5%	86.3 ± 5.4%	94.9 ± 3.4%	94.0 ± 4.2%

**Table 4 sensors-22-07960-t004:** Confusion matrix table result of CNN.

CNN		Predicted Gait Pattern			
		NG	PO	KH	Precision
Actual gait pattern	NG	695	1	4	99.3%
	PO	2	661	37	94.4%
	KH	8	29	663	94.7%
Recall		98.6%	95.7%	94.2%	

**Table 5 sensors-22-07960-t005:** Confusion matrix table result of LSTM.

LSTM		Predicted Gait Pattern			
		NG	PO	KH	Precision
Actual gait pattern	NG	690	2	8	98.6%
	PO	4	614	82	87.7%
	KH	5	49	646	92.3%
Recall		98.7%	92.3%	87.8%	

**Table 6 sensors-22-07960-t006:** Confusion matrix table result of SVM.

SVM		Predicted Gait Pattern			
		NG	PO	KH	Precision
Actual gait pattern	NG	691	4	5	98.7%
	PO	4	659	37	94.1%
	KH	5	32	663	94.7%
Recall		98.7%	94.8%	94.0%	

**Table 7 sensors-22-07960-t007:** Confusion matrix table result of KNN.

KNN		Predicted Gait Pattern			
		NG	PO	KH	Precision
Actual gait pattern	NG	681	11	8	97.3%
	PO	3	661	36	94.4%
	KH	5	55	640	91.4%
Recall		98.8%	90.9%	93.6%	

**Table 8 sensors-22-07960-t008:** Accuracy of Individual ML algorithms for different Gait Recognition patterns.

ML	Methods	PO Gait		KH Gait		Normal Gait	
Correct Prediction		No	Yes	No	Yes	No	Yes
SVM	Count	36 (5.2%)	659 (94.8%)	42 (6.0%)	663 (94.0%)	9 (1.3%)	691 (98.7%)
	*p* Value	0.089	0.089	0.057	0.057	0.993	0.993
KNN	Count	66 (9.1%)	661 (90.9%)	42 (6.2%)	640 (93.8%)	8 (1.2%)	681 (98.8%)
	*p* Value	0.001	0.001	0.102	0.102	0.732	0.732
CNN	Count	30 (4.3%)	661 (95.7%)	41 (5.8%)	663 (94.2%)	10 (1.4%)	695 (98.6%)
	*p* Value	0.007	0.007	0.040	0.040	0.724	0.724
LSTM	Count	51 (7.7%)	614 (92.3%)	90 (12.2%)	646 (87.8%)	9 (1.3%)	690 (98.7%)
	*p* Value	0.194	0.194	0.000	0.000	0.997	0.997
Chi-square, Bonferroni correction test							

## Data Availability

Not applicable.

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
