# Peer review of "Computer Vision and Machine Learning-Based Gait Pattern Recognition for Flat Fall Prediction"

_sensors, 2022, doi:10.3390/s22207960_

Round 1

Reviewer 1 Report

This paper is well written and organized, and there are only some small suggestions.

(1) Technical details should be added to the paper, including the specific structures of CNN and LSTM, e.g. step size and kernel size.

(2) Although the authors have compared various evaluation indicators, the reasoning speed of the model is still encouraged to be reported.

(3) Hyperparameter settings and optimization methods for model objective function optimization should be introduced. Whether the learning rate is fixed or annealed?

(4) The key point of LSTM or CNN models is their well generalization ability across different scenes. Thus, authors should analyze or discuss the regularization and generalization techniques of LSTM and CNN in the Related Work, such as “Improvement of generalization ability of deep CNN via implicit regularization in two-stage training process,” IEEE Access, vol. 6, pp. 15844-15869, 2018. “Cloud shape classification system based on multi-channel cnn and improved fdm.” IEEE Access 8 (2020): 44111-44124. “GPU-accelerated faster mean shift with Euclidean distance metrics.” IEEE 46th Annual Computers, Software, and Applications Conference (COMPSAC). IEEE, 2022. “Pseudo RGB-face recognition,” IEEE Sensors Journal, 2022, doi: 10.1109/JSEN.2022.3197235.

Author Response

The authors greatly appreciate the reviewers’ valuable comments and suggestions. The followings are the author’s response to the reviewers’ comments.

Comment: Technical details should be added to the paper, including the specific structures of CNN and LSTM, e.g. step size and kernel size.

Answer: According to the reviewer’s comments, we have added two block diagrams to describe the specific structures of the proposed CNN (Figure 3) and LSTM (Figure 4). Detailed information about CNN was added in the manuscript in lines 58-270, and lines311-320 for LSTM.

Comment: Although the authors have compared various evaluation indicators, the reasoning speed of the model is still encouraged to be reported.

Answer: According to the reviewer’s comments, we compared the runtime of four proposed models in table 2 to show the reasoning speed of the models. And we also discussed it in lines403-411.

Comment: Hyperparameter settings and optimization methods for model objective function optimization should be introduced. Whether the learning rate is fixed or annealed?

Answer: According to the reviewer’s comments, we have added the description of the hyperparameter settings and optimization methods in lines 2 277 for CNN, and lines 7-320 for LSTM.

Comment: The key point of LSTM or CNN models is their good generalization ability across different scenes. Thus, authors should analyze or discuss the regularization and generalization techniques of LSTM and CNN in the Related Work, such as “Improvement of the generalization ability of deep CNN via implicit regularization in the two-stage training process,” IEEE Access, vol. 6, pp. 15844-15869, 2018. “Cloud shape classification system based on multi-channel CNN and improved FDM.” IEEE Access 8 (2020): 44111-44124. “GPU-accelerated faster mean shift with Euclidean distance metrics.” IEEE 46th Annual Computers, Software, and Applications Conference (COMPSAC). IEEE, 2022. “Pseudo RGB-face recognition,” IEEE Sensors Journal, 2022, doi: 10.1109/JSEN.2022.3197235.

Answer: According to the reviewer’s comments, we have discussed more the CNN and LSTM in related work. The suggested references have been cited in lines 247, and line298.

Reviewer 2 Report

The main contribution and proposed approach have some novelty in contribution. Revision in terms of technical details is needed before publication. So, some comments are suggested to describe technical details.

1. Four different classifiers are applied in your proposed approach. So, it is suggested to discuss about the runtime. (Comparing with existing methods is not necessary)

2. Add a block-diagram of your proposed approach with related main steps in the text.

3. KNN can be performed with different K values. Add more details about K in your experiments.

4. CNN is used in this paper for classification phase. Discuss about your designed CNN with more details about the layers, parameters, etc.

5. Your proposed approach can be used in medical applications for motion tracking. Also, it can be used in image retrieval too. For example, I find a paper titled “Elevation of Some Biochemical and Immunological Parameters in Hemodialysis Patients Suffering from Hepatitis C Virus Infection in Babylon Province”, which has relation. Also, the paper titled “Content based image retrieval based on weighted fusion of texture and color features derived from modified local binary patterns and local neighborhood difference patterns”, has enough relation. Cite these papers and discuss about it as one of the advantages of your proposed method.

6. Is your proposed approach sensitive to input image size and the color space? Discuss briefly about it.

Author Response

The authors greatly appreciate the reviewers’ valuable comments and suggestions. The followings are the author’s responses to the reviewers’ comments.

Comment: Four different classifiers are applied in your proposed approach. So, it is suggested to discuss about the runtime. (Comparing with existing methods is not necessary)

Answer: According to the reviewer’s comments, we have discussed about the runtime of the proposed method in lines403-411. And Table 2 shows the detailed runtime of the models.

Comment: Add a block diagram of your proposed approach with related main steps in the text.

Answer: According to the reviewer’s comments, we have added the block diagram for CNN (Figure 3) and LSTM (Figure 4).

Comment: KNN can be performed with different K values. Add more details about K in your experiments.

Answer: According to the reviewer’s comments, we have discussed more about K values of the proposed KNN in lines 300-302. Also, the best K values of models were shown in table 2 and lines 410

Comment: CNN is used in this paper for the classification phase. Discuss your designed CNN with more details about the layers, parameters, etc.

Answer: According to the reviewer’s comments, we have added more detailed information about the proposed CNN in lines 258-275

Comment: Your proposed approach can be used in medical applications for motion tracking. Also, it can be used in image retrieval too. For example, I find a paper titled “Elevation of Some Biochemical and Immunological Parameters in Hemodialysis Patients Suffering from Hepatitis C Virus Infection in Babylon Province”,

Answer: This suggested paper is not related to any image analysis/ processing, nor motion capture, hence it has not been cited in our paper.

Comment: Also, the paper titled “Content based image retrieval based on weighted fusion of texture and color features derived from modified local binary patterns and local neighborhood difference patterns[1]”, has enough relation. Cite these papers and discuss it as one of the advantages of your proposed method.

Answer: According to the reviewer’s comments, we have discussed more applications of the proposed method in related work. The suggested reference is cited in line 436

Comment: Is your proposed approach sensitive to input image size and color space? Discuss briefly about it.

Answer: It is not clear if the image size or the color space. It has been reported in an outdoor environment[2], data accuracy can be affected. Image size and color space can influence joint reading accuracy upon using computer cameras.[3] The stationary environment and experimental setting did significantly change image size and color space between different participants, the image size or color space did not affect our joint angle readings.

Answer: According to the reviewer’s comments, the references have been cited.

  1. Kayhan, N.; Fekri-Ershad, S. Content based image retrieval based on weighted fusion of texture and color features derived from modified local binary patterns and local neighborhood difference patterns. Multimedia Tools and Applications 2021, 80, 32763-32790, doi:10.1007/s11042-021-11217-z.
  2. Hu, W.; Chi, J.; Liu, J.; Yang, Z. A Novel Method for Space Circular Target Detection in Machine Vision. Sensors (Basel) 2022, 22, doi:10.3390/s22030769.
  3. Kim, H.-K.; Park, J.H.; Jung, H.-Y. An Efficient Color Space for Deep-Learning Based Traffic Light Recognition. Journal of Advanced Transportation 2018, 2018, 1-12, doi:10.1155/2018/2365414.

Round 2

Reviewer 2 Report

The revised version is improved in terms of paper organization and technical details. Most of comments are considered in the text. The proposed approach has enough novelty in methodology.